

# Constraining Uncertainties in CMIP5 Projections of Arctic Sea Ice Volume with Observations

Wang Yangjun, Liu Kefeng, Shang Yulong, Zhang Ren

College of Meteorology and Oceanography, National University of Defense Technology, Nanjing, 211101, China.

*Correspondence:* Liu Kefeng (lkfnudt@sina.com*)*

**Abstract.** This study proposes adaptive forecasting through exponential re-weighting based on the Structural Similarity Index Measure (AFTER-SSIM) algorithm to evaluate the performance of global climate models from the Coupled Model Intercomparison Project (CMIP5) under different emission scenarios during 2006 to 2018, attempting to reduce the uncertainty among them. The SSIM approach uses a loss function to obtain more

information on the spatial distribution between model outputs and observed data, where the genetic algorithm (GA) is used to optimise the parameters of both seasonal cycles and long-term trends of sea ice concentration and sea ice thickness. The re-weighting mechanism of the AFTER-SSIM algorithm guarantees a performance improvement in sea ice volume simulations as new information is added. Finally, the ranked models have been combined to estimate the future Arctic sea ice volume and navigation possibility through the Arctic Northern Sea

Route. Results show that the proposed algorithm reduces the uncertainty among models, sea ice volume will continue to shrink in the future, and the open periods for 1A super vessels are likely to reach to five months ranging from August to December in 2030.

## 1 Introduction

In a warming climate, the sea ice extent of the Arctic region has shown a consistent decrease, ranging between

3.5% and 4.1% every decade from 1979 to 2012 (Stocker et al. 2013). This has contributed to the extended open duration of the Arctic routes and drawn international attention to energy exploration, shipping industry, and its regional ecosystem. The focus of research has shifted from sea-ice loss trends to the open period of the Arctic water (Wang & Overland, 2015).

Global climate models (GCMs) provided from the Coupled Model Intercomparison Project (CMIP5, Taylor et al.

2012) are currently prevalent in the projection of future sea-ice conditions as the most advanced climate models available for the scientific cycles, showing a continued shrinking and thinning in the sea ice in all future scenarios (Massonnet et al. 2012).

However, large uncertainty exists in the current GCMs for future sea ice projections, caused by the discrepancies in initial conditions or physical processes in the ocean or atmospheric simulation as well as the differences in grid

resolution ( Hawkins & Sutton, 2009; Stroeve et al., 2012; Wang & Overland, 2012; Swart et al., 2015; Wu et al.,



2018), which block the accurate estimate for the open duration of Arctic routes. Despite these uncertainties, GCMs are powerful tools to understand the future changes in Arctic sea ice. Wang & Overland, (2009) pointed out that these models could be applied in projections with careful evaluation. Wang & Overland, (2015) have reduced 37 GCMs to 12 models by taking both the mean trends and seasonal cycles of the September sea ice extent (SSIE)

projections from 1981 to 2005 into consideration.

Since 2006, four future scenarios called the Representative Concentration Pathways (RCPs) have been used in all GCMs and defined by their total radiative forcing (a cumulative measure of human emissions of greenhouse gas), i.e. RCP26, RCP45, RCP60, and RCP85 (van Vuuren et al. 2011). The number of realisations of these 12 GCMs extends to 101, leading to new uncertainty. Therefore, this study tries to further access the model performance of

101 candidate realisations from 2006 to 2018 and find a reasonable way to reduce the uncertainty existing in the selected models under different emission scenarios.

Generally, there are two ways to improve the precision of projections; one is to find the model that best fits reality, while the other is to combine estimates with multi-models. In the 1970s, these combined methods were doubted by some researchers due to their lack of theoretical justification (Newbold and Granger 1974). Another argument

is that a single best model can be found based on all the information provided by these individual forecasters instead of combining them. However, views have changed over the past 20 years and optimal estimates often face the danger of overfitting (e.g. regression) and lead to high instability. Thus, building such a model is not usually feasible (Yang 2001). Combining estimates is a constructive way that helps to reduce the variability (Chen and Yang 2007). Additionally, few studies have focused on the spatial distribution performance of models, which has

been widely evaluated in other meteorological elements (Basharin et al. 2016; Shi et al. 2017). Additionally, the error method (e.g. relative error, root mean square error), widely used in previous literature (Wang and Overland 2015a, 2009, 2012; Massonnet et al. 2011; Stroeve et al. 2012; Liu et al. 2013), cannot suitably fit the difference in spatial distribution (Zhou et al. 2004). Therefore, a combined method called the adaptive forecasting through exponential re-weighting (AFTER) algorithm is incorporated in this study to evaluate the performance of GCMs

and reduce the uncertainty in future sea-ice projections. The Structural Similarity Index Measure (SSIM) algorithm is introduced to study the similarity of spatial distribution between model output and observed data in both seasonal cycles and mean states. To improve the explanation of physical mechanisms (Shu et al. 2015), sea ice thickness is also taken into account in the calculations.

The rest of this manuscript is organised as follows: Section 2 introduces the data used in this study and their features compared with the previous study. The AFTER-SSIM approach, the model ranking workflow, and future sea ice projections are introduced in Section 3. Section 4 shows the results of the model evaluation and combined estimates of future sea ice volume; consequently, the open period of the Arctic water can also be obtained.

## 2 Data description

### 2.1 PIOMAS data

Spatial consistency, temporal length, and completeness are key factors in data evaluation (Melia et al. 2015). Therefore, Pan-Arctic Ice-Ocean Modelling and Assimilation System (PIOMAS) sea-ice reanalysis data are used to estimate the observed sea ice condition in this study (Zhang and Rothrock 2003). Despite the uncertainty in the PIOMAS data, the current observation values (i.e. ICEsat or CryoSat-2) have limited spatial coverage and temporal scale, which restrict the model evaluation ability. Large uncertainty and bias still exist in the inversion methods of

satellites for driving factors (i.e. sea ice concentration (SIC), sea ice age, and snow depth). The sea ice thickness of the PIOMAS data was observed to fit well with the observations (ICESat field) with less than a 0.1 mean difference and high pattern correlation ($r > 0.8$). The spatial patterns, seasonal cycles, and trends in sea ice thickness (SIT) are realistically reproduced due to atmospheric reanalysis forcings (Labe et al. 2018). Therefore, PIOMAS data have been widely used to represent observations in several studies (Shu et al. 2015; Labe et al.

2018). For temporal consistency, SIC and SIT data were provided by PIOMAS in this study.

### 2.2 Multi-model data

Twelve GCMs, identified by Wang & Overland, (2015), are used for further evaluation: ACCESS1.0, ACCESS1.3, CCSM4, CESM1(CAM5.1), EC-EARTH, HadGEM2-AO, HadGEM2-CC, HadGEM2-ES, MIROC-ESM, MIROC-ESM-CHEM, MPI-ESM-LR, and MPI-ESM-MR. There are 101 ensemble candidates from these 12

models in all emission scenarios. SIC and SIT data are derived from the 12 CMIP5 models to compare with the PIOMAS data. Table 1 presents the basic characteristics of the selected GCMs, where each model has different spatial resolution and ensemble members.




### 2.3 Data processing

Data from GCMs and PIOMAS with different resolutions are interpolated into the same $1° \times 1°$ resolution. The

monthly data, $X$ are divided into two parts: the seasonal cycle, $\bar{X}$ and the long-term trend with anomalies, $\hat{X}$ as

follows:

$$X = \bar{X} + \hat{X}, \tag{1}$$

### 2.4 Spatial variation of sea ice

The spatial variation of sea ice between the current mean state (2006–2018) and historical mean state (1979–2005)

in March and September, respectively, derived from the PIOMAS data can be seen in Figure 1. Both in March and

September, the coverage of sea ice has shrunk compared to the historical period; the decrease in September is more

evident than that in March. In March, the reduction of the mean SIC mainly occurs in the Sea of Okhotsk, Barents

Sea, and part of the Greenland Sea; in September, the reduction area contains the Beaufort Sea, Chukchi Sea, East

Siberian Sea, and Laptev Sea, which extends to $80°N$.

The current mean SIT has thinned down compared to the historical mean SIT. Most of the Arctic area shows a

larger reduction of SIT in September than in March. The SIT throughout the Arctic area is no more than 2 meters.

For these two months, the variation of mean SIT is more informative than that of mean SIC, especially in the

central area, where sea ice thinning can occur without major variations in the local SIC (Melia et al. 2015).

Considering the changes in the spatial distribution for both the mean SIC and mean SIT, the sea ice conditions

have varied much. The Northern Sea Routes along the coast of the Chukchi Sea, East Siberian Sea, Laptev Sea,

Kara Sea, and Barents Sea begin to be ice-free in September during the current period, as well as the Northwest

Passages along the coast of the Beaufort Sea. Therefore, it is necessary to further evaluate model performance in

the new period.

### 2.5 Temporal variation of sea ice

Sea ice volume (SIV), which takes both SIC and SIT into consideration, is a good index for the evaluation of

model performance (Shu et al. 2015). SIV can be calculated as the sum of the grid cell area of, that is, the SIC and

SIT of each grid cell, which can be represented as follows:

$$SIV = \sum_{lon=-180}^{180} \sum_{lat=66.5}^{90} SIT(lon, lat) \cdot SIC(lon, lat) \cdot 2\pi r^2 \cdot \left( sin\left(\frac{lat+1}{180} \cdot \pi\right) - sin\left(\frac{lat}{180} \cdot \pi\right) \right)/360, \tag{2}$$





where $r$ is radius of the earth; $lon$ and $lat$ represent the longitude and latitude of the gird, respectively.

Note that although the SIV values from the PIOMAS data are model simulations with data assimilation, they can competently assess the GCMs' performances. The change in monthly average SIV between the historical period and current period can be seen in Figure 2. Compared to the historical period, the SIV has suffered shrunk every month during the current period. In spring, the SIV reaches its peak in April at $29.5 \times 10^3 km^3$ in the historical mean (1979-2005) and at $23.4 \times 10^3 km^3$ in the current mean (2006-2018). The smallest SIV appears in

September at $13.3 \times 10^3 km^3$ in the historical mean and at $5.8 \times 10^3 km^3$ in the current mean. Additionally, SIV reaches its minimum in September 2012 at $3.79 \times 10^3 km^3$, which might have been caused by an unusually strong storm in the central Arctic basin from April to August 2012 (Parkinson and Comiso 2013).

Compared to the period of 1979–2005, the estimated negative trend in SIV is approximately -35.4% from 2006–2018. Additionally, during most of the 2006–2018 period, SIV shows a downward trend with three rebounds in

2008, from 2013–2014, and in 2018. Swart et al. (2015) posited that sea ice change is driven by external forcing and internal variability. SIV rebounds when the effect of the internal variability masks the external forcing. Overall, SIV shows much new information in the period 2006–2018 and is worth studying further.

## 3 Methods

### 3.1 Structural Similarity Index Measure (SSIM)

Wang et al. (2004) proposed an objective method for accessing the structural similarity between two images. This method has been widely used in measuring the image quality. Compared to the traditional error methods, this new method termed as the Structural Similarity Index Measure (SSIM) and can better depict the difference in the spatial distribution between two data sets. The formula of the SSIM can be presented as follows:

$$SSIM(X_{mod}, X_{obs}) = l(X_{mod}, X_{obs})^\alpha \cdot c(X_{mod}, X_{obs})^\beta \cdot s(X_{mod}, X_{obs})^\gamma, \tag{3}$$

where $l(X_{mod}, X_{obs})$ represents the variation of mean value, $c(X_{mod}, X_{obs})$ is the variation of deviation, and $s(X_{mod}, X_{obs})$ stands for the structure variation, which can be presented as follows:

$$l(X_{mod}, X_{obs}) = \frac{\left(2\mu_{x_{mod}}\mu_{x_{obs}} + C_1\right)}{\left(\mu^2_{x_{mod}} + \mu^2_{x_{obs}} + C_1\right)}, \tag{4}$$

$$c(X_{mod}, X_{obs}) = \frac{2\sigma_{x_{mod}}\sigma_{x_{obs}} + C_2}{\sigma^2_{x_{mod}} + \sigma^2_{x_{obs}} + C_2}, \tag{5}$$




$$s(X_{mod}, X_{obs}) = \frac{\sigma_{x_{mod}x_{obs}} + C_3}{\sigma_{x_{mod}}\sigma_{x_{obs}} + C_3}, \tag{6}$$

where $\mu$ is the mean value of $X$, $\sigma$ is the standard deviation of $X$, $C_1, C_2, C_3$ are constants to avoid the system

unstability. If we suppose $\alpha = \beta = \gamma = 1, C_3 = C_2/2$, then the Eq. (3) can be rewritten as follows:

$$SSIM(X_{mod}, X_{obs}) = \frac{\left(2\mu_{x_{mod}}\mu_{x_{obs}} + C_1\right)\left(2\sigma_{x_{mod}x_{obs}} + C_2\right)}{\left(\mu^2_{x_{mod}} + \mu^2_{x_{obs}} + C_1\right)\left(\sigma^2_{x_{mod}} + \sigma^2_{x_{obs}} + C_2\right)}, \tag{7}$$

Let us assume a random matrix $A$ of $10 \times 10$, ranging from 0 to 1, If 10 is added to the last element, we can

obtain matrix $B$. If we add 1 or -1 randomly to each element, matrix $C$ can be obtained. We compare matrix $B$

and $C$ with the original matrix $A$ respectively; the root mean square error (RMSE) of these two are the same,

while the SSIM of matrix B and A is larger than that of matrix C and A, showing that matrix $B$ is structurally

more similar to matrix A than matrix $C$ (see Figure 3). This example presents the advantage of SSIM over the

RMSE for spatial distribution analysis.

Using this method, we evaluate the structural similarity of the seasonal cycles and the long-term trends in SIC

between the GCMs and PIOMAS data, as well as that in SIT. The four scores can be written as follows:

$$SC_j^{sic} = \frac{1}{M}\sum_{k=1}^{M} SSIM(\bar{X}^{sic}_{mod_{jm}}, \bar{X}^{sic}_{pio_m}), \tag{8}$$

$$SC_j^{sit} = \frac{1}{M}\sum_{k=1}^{M} SSIM(\bar{X}^{sit}_{mod_{jm}}, \bar{X}^{sit}_{pio_m}), \tag{9}$$

$$ST_j^{sic} = \frac{1}{M}\sum_{k=1}^{M} SSIM(\hat{X}^{sic}_{mod_{jm}}, \hat{X}^{sic}_{pio_m}), \tag{10}$$

$$ST_j^{sit} = \frac{1}{M}\sum_{k=1}^{M} SSIM(\hat{X}^{sit}_{mod_{jm}}, \hat{X}^{sit}_{pio_m}), \tag{11}$$

where $SC_j^{sic}$ and $SC_j^{sit}$ are the similarities in the seasonal cycles of SIC (CSIC) and SIT (CSIT) between the $j$th

ensemble member of GCMs and PIOMAS data, respectively; $ST_j^{sic}$ and $ST_j^{sit}$ are the similarities in the long-

term trends of SIC (TSIC) and SIT (TSIT) between the $j$th ensemble member of GCMs and PIOMAS data,

respectively; and $M$ is the number of the time series.



**3.2 AFTER-SSIM method**

Yan (2001) first proposed the scheme of the AFTER algorithm to develop combinations for better forecasts. Simulations and real data examples have shown the advantages and applications of the AFTER algorithm (Altavilla and Grauwe 2010; Rapach and Strauss 2008; Sánchez 2008; Shu et al. 2009), which is defined as:

$$W_{i,j} = \frac{\prod_{k=1}^{i-1} \hat{s}_{k,j}^{-1/2} \exp\left(-\lambda L\left(\frac{Y_k - \hat{y}_{k,j}}{\hat{s}_{k,j}}\right)\right)}{\sum_{j'=1}^{J} \prod_{k=1}^{i-1} \hat{s}_{k,j}^{-1/2} \exp\left(-\lambda L\left(\frac{Y_k - \hat{y}_{k,j}}{\hat{s}_{k,j}}\right)\right)}, \tag{12}$$

where $W_{i,j}$ is the weight of each model $j \in \Theta$ at each time $i \in$ I; note that $\sum_{j=1}^{\infty} W_{i,j} = 1$ for $i \geq 1$, $\lambda$ is a tuning

parameter to control the degree of weighting dependence on the predictive performance (Wei and Yang 2011). The L1 loss function used in this manuscript can be written as follows:

$$L(\frac{Y_k - \hat{y}_{k,j}}{\hat{s}_{k,j}}) = \left|\frac{Y_k - \hat{y}_{k,j}}{\hat{s}_{k,j}}\right|, \tag{13}$$

The variance $\hat{s}_{k,j}$ can be estimated as follows:

$$\hat{s}_{k,j} = \frac{1}{i-1} \sum_{k=1}^{i-1} |Y_k - \hat{y}_{k,j}|, \tag{14}$$

Generally, relative error is used in majority of the studies to distribute the weights of candidate models (e.g. Wei and Yang 2012; Yang 2001), where $Y_k$ represents the PIOMAS data at time $K$, while $\hat{y}_{k,j}$ represents the $j$th model data at time $K$. In this study, instead of relative error, a new form has been established to represent the difference between the model outputs and PIOMAS data, which can be presented as follows:

$$Y_k - \hat{y}_{k,j} = exp\left(-2 \cdot \left(\alpha_1 SC_j^{sic} + \alpha_2 SC_j^{sit} + \alpha_3 ST_j^{sic} + \alpha_4 SC_j^{sic}\right)\right), \tag{15}$$

where $SC_j^{sic}, SC_j^{sit}, ST_j^{sic}, SC_j^{sic}, j \in \Theta$ are derived from SSIM, $\alpha$ is the weight vector, and $\sum_{m=1}^{4} \alpha_m = 1$. Note that, the weight factor in this study has been optimised by the genetic algorithm (GA) (Whitley 1994).

The ensemble forecast procedure $\hat{y}_i^*$ can be represented as:

$$\hat{y}_i^* = \sum_{j \in \Theta} W_{i,j} \, \hat{y}_{i,j}, \tag{16}$$

where $i$ is the projection time.

If we rewrite Eq. (11), a form closely related to the Bayesian update can be found:




$$W_{i,j} = \frac{W_{i-1,j} \exp\left(-\lambda L\left(\frac{Y_k - \hat{y}_{k,j}}{\hat{s}_{k,j}}\right)\right)}{\sum_{j'} W_{i-1,j'} \exp\left(-\lambda L\left(\frac{Y_k - \hat{y}_{k,j}}{\hat{s}_{k,j}}\right)\right)}, \tag{17}$$

After each additional observation, the weights of each model can be updated. Thus, we call this algorithm adaptive forecasting through an exponential re-weighting method. Specifically, the common AFTER algorithm is termed as AFTER-RE and the new form AFTER algorithm is termed as AFTER-SSIM.

### 3.3 Navigability of the Arctic routes

The navigability of the Arctic Routes can be represented by the Ice Numeral (IN) index derived from the Arctic Ice Regime Shipping System (AIRSS), where both SIC and SIT are taken into consideration (Howell and Yackel 2004; Smith and Stephenson 2013; Stephenson and Smith 2015; CanadaTransport 1998). The IN is given by

$$IN = C_a IM_a + C_b IM_b + \cdots + C_n IM_n, \tag{18}$$

where $C_n$ is the concentration in tenths of ice type $n$ and $IM_n$ is the Ice Multiplier for ice type $n$. Ice-type describes the specific stage of ice development, which is closely related to the ice age and thickness. Ice Multipliers (a series of integers), determined by ship class and ice type, are used to illustrate the impact of sea ice type on a specific vessel. $IM < 0$ reflects the ice obstacle effects on vessels. Ice types are determined by CanadaTransport, (1998) and Johnston, (2017). Note that the area can be navigable only if the IN index is larger than zero. Details regarding the ice type and Ice Multiplier can be seen in Tables 2 and 3.

### 3.4 Work flow

The workflow of the proposed method can be seen in Figure 4. In Step 1, the PIOMAS data and model are interpolated into the same $1° \times 1°$ resolution, and the original data are separated into the seasonal cycles and long-term trends with anomalies. In Step 2, the seasonal cycle in the model data is compared with that in the PIOMAS data by the SSIM method to reflect the model's reaction to the seasonal variation of the solar cycle. The SSIM method is also used to calculate the long-term trends between multi-models and PIOMAS to reflect the model's fidelity to the real world. In Step 3, we proposed an AFTER-SSIM algorithm to calculate the weights for every model and generate an ensemble forecast for the future sea ice volume (SIV), where the seasonal cycle scores (CSIC/CSIT), as well as the long-term trend scores (TSIC/TSIT), are incorporated into the loss function and variance estimation. The parameters in the AFTER-SSIM algorithm can be optimised by GA. Additionally, the


candidate realisations are ranked in terms of their weights. In Step 4, the calculated weights are used to modify the
SIC and SIT fields; then, the future open period of the Arctic sea routes can be obtained.

**4 Results and Discussion**

**4.1 Scores from SSIM between GCMs and PIOMAS data**

A total of 101 candidate members from the 12 models were selected by Wang & Overland, (2015) in all emission

scenarios. Based on the workflow proposed in Section 3.1, we can obtain four kinds of SSIM scores for each
ensemble model (see Figure 5). From Figure 5, all the realisations have better performances in the simulation of
seasonal cycles than in long-term trends. The average scores of the seasonal cycles between all GCMs and
PIOMAS data can reach 0.7717 for SIC and 0.7427 for SIT. HadGEM2-AO, HadGEM2-CC, and HadGEM2-ES
show better performances in the simulation of the seasonal cycles of SIC, while MPI-ESM-LR and MPI-ESM-

MR show the best performances in the simulation of the seasonal cycles of SIT. MIROC-ESM, MIROC-ESM-
CHEM, MPI-ESM-LR, and MPI-ESM-MR have advantages in modelling SIT trends, while MPI-ESM-LR and
MPI-ESM-MR have advantages in simulating SIC trends.

**4.2 Model rank based on sea ice volume**

To further rank the models in terms of their performance in sea ice conditions, we introduce these four SSIM

scores together into the AFTER algorithm as its loss function; their weights can be determined by GA. SIV is used
as an index to measure the model performance of both SIC and SIT. The simulation results can be seen in Table 4
and Figure 6.

Table 4 compares the simulation performance of SIV based on different methods and their stability by computing
the RMSE between GCMs and PIOMAS data. The variable $n$ represents the number of months used in the training

from 2006–2017; all 12 months in 2018 are used to test the model performance. AFTER-SSIM is the proposed
method in this study, AFTER-RE is the AFTER algorithm that uses relative error as the loss function, LR is the
linear regression method, GRNN is the generalised regression neural network, RF represents the random forest
algorithm, Mean is the average outcome of all the models, and Single is the optimal model among the realisations.
From Table 4, RF has the lowest RMSE if the samples are less than 144 but becomes large when the sample

number reaches 144, thus, demonstrating some instability.




The AFTER-SSIM algorithm has the second-lowest RMSE among all the schemes with robust stability. The GRNN algorithm shows good performances in both RMSE and stability, while the model performance of LR deteriorates rapidly as the number of samples decreases.

The difference between AFTER-SSIM and AFTER-RE is the choice of loss functions. The SSIM approach can
determine more information than the RE method, which was discussed in Section 3.1; thus, AFTER-SSIM has a lower RMSE than AFTER-RE. Additionally, the weights of the AFTER-SSIM algorithm can be updated by adding new observations (see Eq. (17)) to steadily improve the accuracy of combined forecasts in SIV, showing advantages over the GRNN and RF algorithms. Overall, AFTER-SSIM is a good tool to obtain combined forecasts.

Then, the weights of candidate realisations can be obtained by the AFTER-SSIM algorithm (see Figure 6). The
top 34 candidate realisations, only accounting for 33% of the total members, contribute 90.7% of the weight in the combined forecast, as listed in Table 5. Considering Figure 6 and Table 5, the candidate realisations from MPI-ESM-LR have the largest weights (41.27%), followed by candidate realisations from MPI-ESM-MR (11.18%), CCSM4 (24.73%), MIROC-ESM (6.99%), and MIROC-ESM-CHEM (6.53%), which are key factors in the combined projection. Models from ACCESS1.0, ACCESS1.3, CESM4, EC-EARTH, HADGEM2-AO,
HADGEM2-CC, and HADGEM2-ES have relative low weights and contribute less to the combined projection.

Then, Shannon's entropy is used as a tool to measure the variation of uncertainty between the initial candidate realisations and ranked models (Shannon, 1948). The formula can be written as follows:

$$U_k = \sum_{j=1}^{3} -p_k ln(p_k), \text{k=1, 2,} \tag{19}$$

where $U_1$ is the uncertainty of original candidate realisations, $U_2$ is the uncertainty of the ranked candidate
realisations, and $p_k, k = 1, 2$ represents the possibility of each model. For the original candidate realisations, the possibility $p_1$ for each model is equal to 1/101, while for the ranked models the possibility $p_2$ for each model is substituted by the weights derived from the AFTER-SSIM algorithm. Hence, the uncertainty is reduced from 4.6152($ln101$) to 3.9061 ($\approx ln50$) by the AFTER-SSIM algorithm, indicating that the information originally scattered in the 101 candidate realisations has been concentrated into approximately 50 models.

Instead of selecting the models, all the candidate realisations are combined with the obtained weights of the historical data reconstruction and future projection for SIV (see Figure 7). In Figure 7, the reconstruction of SIV by the AFTER-SSIM algorithm, based on the optimal loss function (red line), fits the PIOMAS data (black line) well, except for the simulation of some extreme values (e.g. the high values in 2006, 2008–2009 and 2015 and the



low values in 2010–2013 and 2016–2017). This issue can be explained by the predictability of CMIP5 models and

reliability of observations. For CMIP5 models, most large-scale physical sea ice processes, including basic

thermodynamic and dynamic changes, have been well understood and represented (Hunke et al. 2011). However,

some details in the small-scale sea ice dynamic process and mechanical deformation require closer examination

(Girard et al. 2009; Hutchings et al. 2011). Some snow processes such as wind redistribution, vapour transport,

and snow particle changes are not contained in the models (Lecomte et al. 2011). Studies show that the internal

variability of sea ice has accounted for 30–50% of the total observed sea ice change since 1979 (Ding et al. 2017).

The sensitivity of sea ice to atmospheric circulation changes in CMIP5 models is lower than what has been

observed (Rosenblum and Eisenman 2017). For example, research has suggested that the record low SIV in 2012

may have been caused by an unusually strong storm in the central Arctic basin from April to August 2012

(Parkinson and Comiso 2013), which cannot be represented well by CMIP5 models. This has made the

reconstructed SIV appear higher than PIOMAS data. Regarding observation precision, studies have shown that

the satellite retrieval algorithms of SIV often neglect snow thickness changes (Bunzel et al. 2018), as well as

numerous geophysical parameter assumptions (e.g. seawater, snow load, and snow and sea ice densities)

(Zygmuntowska et al. 2014). The current main obstacle to improving the projection of SIV is the lack of long-

term and reliable SIV estimates (Massonnet et al. 2018).

Regardless of the extreme cases, an obvious decrease trend can be seen in the period of 2019–2030 in terms of our

combined forecast. Similar to the 2006–2018 period, the change in SIV in the future will not be a consistent

shrinking as it might rebound in 2020–2021, 2023–2024, and 2028–2029.

### 4.3 Navigability for future Arctic Northern Sea Route

For further exploration of the navigability of the Arctic Northern Sea Route, we combined all the candidate

realisations with their weights to predict the opening period of the Northern Sea Route in 2030. The SIC and SIT

data from 101 candidate realisations are sorted into 11 categories (i.e. $C_n, n = \{0, 1, ..., 10\}$) and 9 categories (i.e.

ice type), respectively. Then the possibility of navigability on each grid can be calculated as follow:

$$P_i = \sum_{j=1}^{N}(C_{ij} \times IM_{ij} \times W_j), \;\; if \; (C_{ij} \times IM_j) > 0, \; i \in \Theta, \tag{20}$$

where $N$ is the number of candidate realisations and $W_j, j = 1, ..., N$ is the weight of each model. The possibility

of navigability for 1A super vessels (the most advanced ice-strengthened vessels) on Northern Sea Route for each

month in 2030 can be seen in Figure 8. It is likely that in the South Barents Sea and Kara Sea, 1A super vessels will be navigable throughout 2030. For the Northern Sea Routes, 1A super vessels are unlikely (less than 60%) to be permitted to sail north of $80°N$ throughout most of 2030. The Laptev Sea, East Siberian Sea, and Chukchi Sea are likely (more than 80%) to be interconnected when August comes, lasting until December. Therefore, the

possibility of 1A super vessels navigating on the Northern Sea Routes for 5 months (from August to December) in 2030 is more than 80%.

### 4.4 Summary

This study proposes a new algorithm called AFTER-SSIM to evaluate the performance of 101 selected global climate models under four different emission scenarios, appearing since 2006.

The SSIM approach is incorporated into the algorithm as a loss function to obtain more information on the spatial distribution between model outputs and PIOMAS data, allowing the AFTER-SSIM algorithm to perform better than the AFTER-RE algorithm. The GA method is used to optimise the parameters in seasonal cycles and long-term trends of SIC and SIT. The re-weighting mechanism of the AFTER-SSIM algorithm ensures the improved performance in SIV simulations as new information is added, showing better performances than other listed

regression methods.

The combined forecasts in SIV with 101 global climate models show that there will be an obvious decrease trend with some rebounds from 2019 to 2030. The invalidity of extreme SIV projection reveals that closer examination is needed for both the physical processes and parameterisations in global climate models as well as to obtain more reliable and long-term observations.

The ranked models are combined to calculate the navigation possibility for 1A super vessels through the Northern Sea Route in 2030. A possibility beyond 80% indicates that the open period can reach five months, ranging from August to December in 2030.

### Acknowledgements

This work is supported by the National Natural Science Foundation of China (41375002). The PIOMAS data set

used in this study is provided by the Polar Science Center and can be accessed at http://psc.apl.uw.edu/research/projects/arctic-sea-ice-volume-anomaly/data/. Data from multi global climate





models including ACCESS1.0, ACCESS1.3, CCSM4, EC-EARTH, CESM1(CAM5.1), HadGEM2-AO, HadGEM2-CC, HadGEM2-ES, MIROC-ESM, MIROC-ESM-CHEM, MPI-ESM-LR, and MPI-ESM-MR used in this study are provided by the fifth phase of the Coupled Model Intercomparison Project (CMIP5), which are

available at https://esgf-node.llnl.gov/search/cmip5/.

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

**Table 1: Basic features of the 12 CMIP5 models used for analysis.**

| Model Number | Model Name | Country | Spatial Resolution | Ensemble members (RCP) 26 | 45 | 60 | 85 | Reference |
|---|---|---|---|---|---|---|---|---|
| 1 | ACCESS1.0 | Australia | tripolar, $1° \times 1°$, refinement at the equator | | 1 | | 1 | (Bi et al. 2013) |
| 2 | ACCESS1.3 | Australia | tripolar, $1° \times 1°$, refinement at the equator | | 1 | | 1 | (Bi et al. 2013) |
| 3 | CCSM4 | USD | dipolar, $1.11° \times (0.27 - 0.54)°$, NP in Greenland | 5 | 6 | 6 | 6 | (Gent and Danabasoglu 2011) |
| 4 | CESM1 | USD | dipolar, $1.11° \times (0.27 - 0.54)°$, NP in Greenland | 3 | | | 1 | (Gent and Danabasoglu 2011) |
| 5 | EC-EARTH | Europe | tripolar, $1° \times 1°$, refinement at the equator | 2 | 10 | | 10 | (Fichefet and Maqueda 1999) |
| 6 | HadGEM2-ES | UK | $(1 - 0.3)° \times 1°$ | 4 | 4 | 4 | 5 | (Mclaren et al. 2006) |
| 7 | HadGEM2-CC | UK | $(1 - 0.3)° \times 1°$ | 1 | 1 | 1 | 1 | (Mclaren et al. 2006) |
| 8 | HadGEM2-AO | Korea | $(1 - 0.3)° \times 1°$ | | 1 | | 3 | (Mclaren et al. 2006) |
| 9 | MIROC-ESM | Japan | $\sim 1.4° \times 1°$ | 1 | 1 | 1 | 2 | (Watanabe et al. 2011) |
| 10 | MIROC-ESM-CHEM | Japan | $\sim 1.4° \times 1°$ | 1 | 1 | 1 | 1 | (Watanabe et al. 2011) |
| 11 | MPI-ESM-LR | Germany | $\sim 1.5° \times 1.5°$ | 3 | 3 | | 3 | (Notz et al. 2013) |
| 12 | MPI-ESM-MR | Germany | $\sim 0.4° \times 0.4°$ | 1 | 3 | | 1 | (Notz et al. 2013) |
| Sum | | | | 21 | 32 | 13 | 35 | |



**Table 2: Ice Type (CanadaTransport, 1998; Johnston, 2017).**

| Ice Type | Characteristic |
|---|---|
| Open Water | Newly formed ice, include ice crystal, grease like ice, crushed ice clusters, etc. These types of ice are loosely frozen together and can only been seen while floating. The ice thickness is less than 10 cm. |
| Grey | Young ice has a thickness of 10–15 cm, which is lower than that of nilas and is easy to expand and break. |
| Grey-white | Young ice has a thickness of 15–30 cm. |
| Thin first year 1st stage | One-year ice, of which the formation time does not exceed one winter, has a thickness of 30–50 cm. |
| Thin first year 2nd stage | One-year ice, of which the formation time does not exceed one winter, has a thickness of 50–70 cm. |
| Medium first year | One-year ice has a thickness of 70–120 cm. |
| Thick first year | One-year ice has a thickness of 120–220 cm. |
| Second year | Adult ice, which has gone through at least one summer melting, has a thickness of 220–250 cm. |
| Multiyear | Multiyear ice, which has gone through at least two summer meltings, has a thickness beyond 250 cm. |

**Table 3: Ice Multiplier for 1A Super (CanadaTransport 1998).**

| | Open Water | Grey Ice | Grey White Ice | Thin First Year 1st Stage | Thin FIRST Year 2nd Stage | Medium First Year | Thick First Year | Second Year | Multi Year |
|---|---|---|---|---|---|---|---|---|---|
| 1A Super | 2 | 2 | 2 | 2 | 2 | 1 | -1 | -3 | -4 |

**Table 4: Comparison for the simulation performance of SIV based on different methods and their stability (n is the number of months used in the training from 2006–2017; all 12 months in 2018 are used to test the model performance)**

| Method/Sample | n=96 | n=108 | n=120 | n=132 | n=144 |
|---|---|---|---|---|---|
| AFTER-SSIM | 0.6323 | 0.6325 | 0.6132 | 0.6083 | 0.6157 |
| AFTER-RE | 2.2619 | 2.1641 | 2.1818 | 2.0157 | 2.0951 |
| LR | 23.3325 | 19.4949 | 8.2884 | 2.5963 | 0.9145 |
| GRNN | 0.8858 | 0.7012 | 0.7682 | 0.8693 | 0.6035 |
| RF | 0.4155 | 0.4526 | 0.3908 | 0.5342 | 0.7478 |
| Mean | 3.6871 | 3.6871 | 3.6871 | 3.6871 | 3.6871 |
| Single member | 1.2802 | 1.2802 | 1.2802 | 1.2802 | 1.2802 |

**Table 5: The top 34 candidate realisations and their weights.**

| Rank | Model | Number of | Weights | Cumulative weight |
|---|---|---|---|---|




|   | Members |   |   |   |
|---|---|---|---|---|
| 1 | MPI-ESM-LR | 9 | 0.4127 | 0.4127 |
| 2 | CCSM4 | 15 | 0.2473 | 0.6600 |
| 3 | MPI-ESM-MR | 4 | 0.1118 | 0.7718 |
| 4 | MIROC-ESM | 3 | 0.0699 | 0.8417 |
| 5 | MIROC-ESM-CHEM | 3 | 0.0653 | 0.9070 |

**Figure 1: Variation of SIC (a) and SIT (b) between the current mean state (2006–2018) and historical mean state (1979–2005) in March and September.**




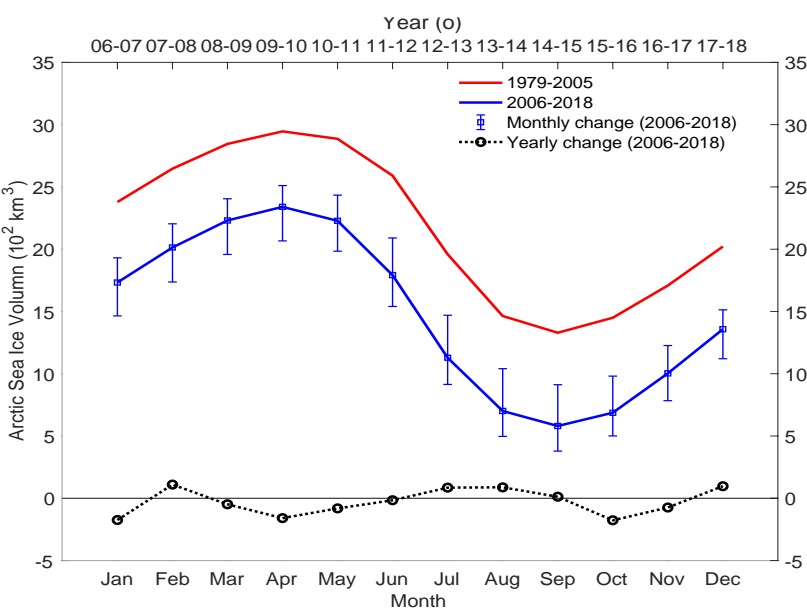

**Figure 2: Variation of the SIV; the red line represents the monthly mean SIV from 1979–2005, and the blue line is the monthly mean SIV from 2006–2018. Compared to the mean state from 2006–2018, the monthly SIV change is represented as a light blue error bar and the yearly change is represented by black dots.**

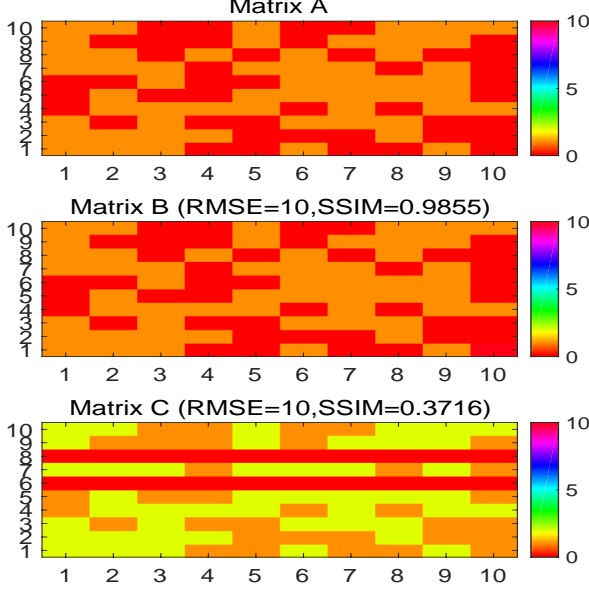


**Figure 3: Example for structure similarity analysis.**


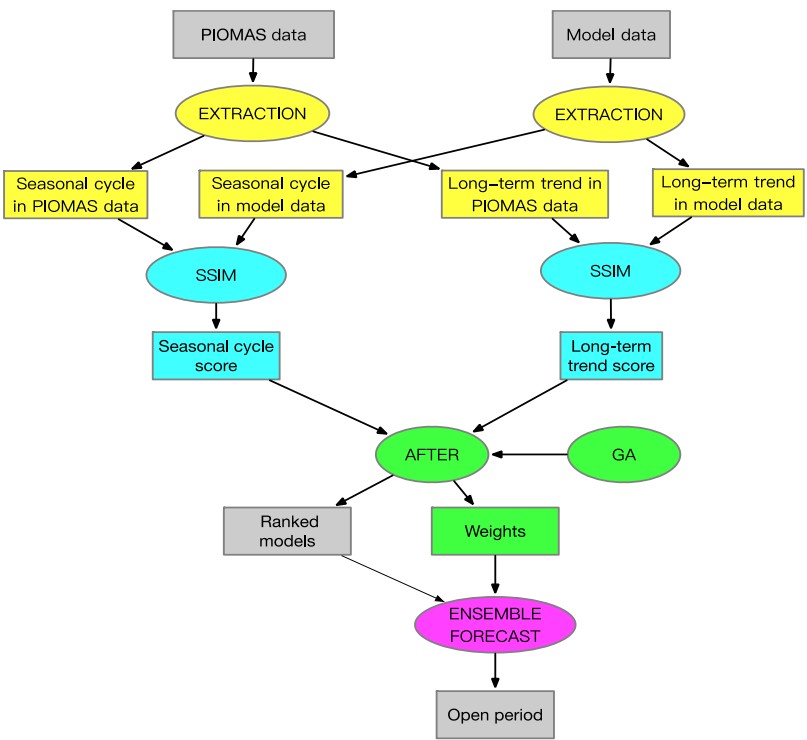

**Figure 4: Work flow of the proposed method.**

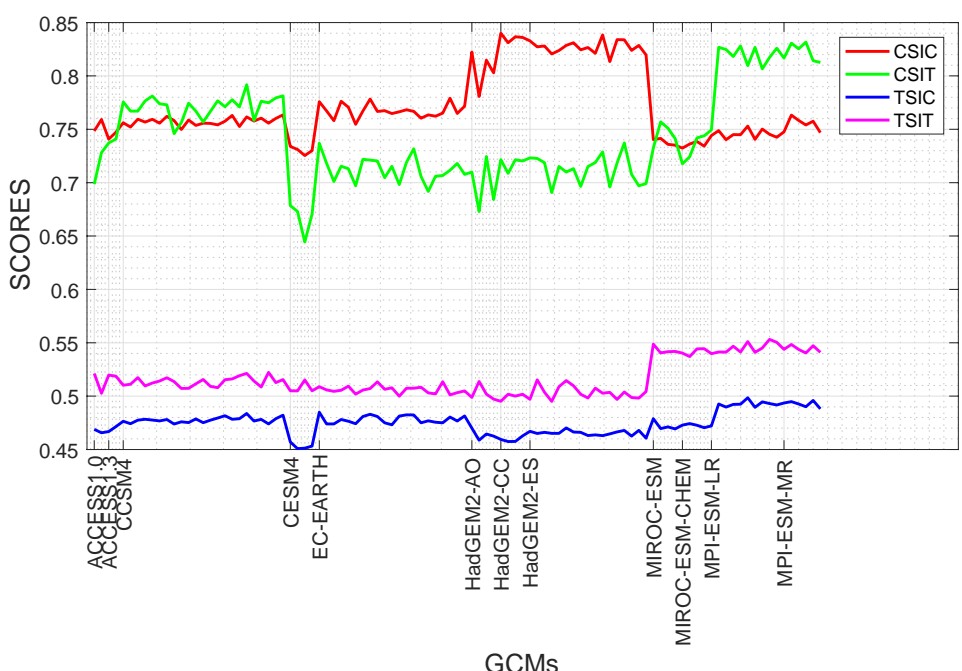




**Figure 5: Model evaluation and selection with four scores.**

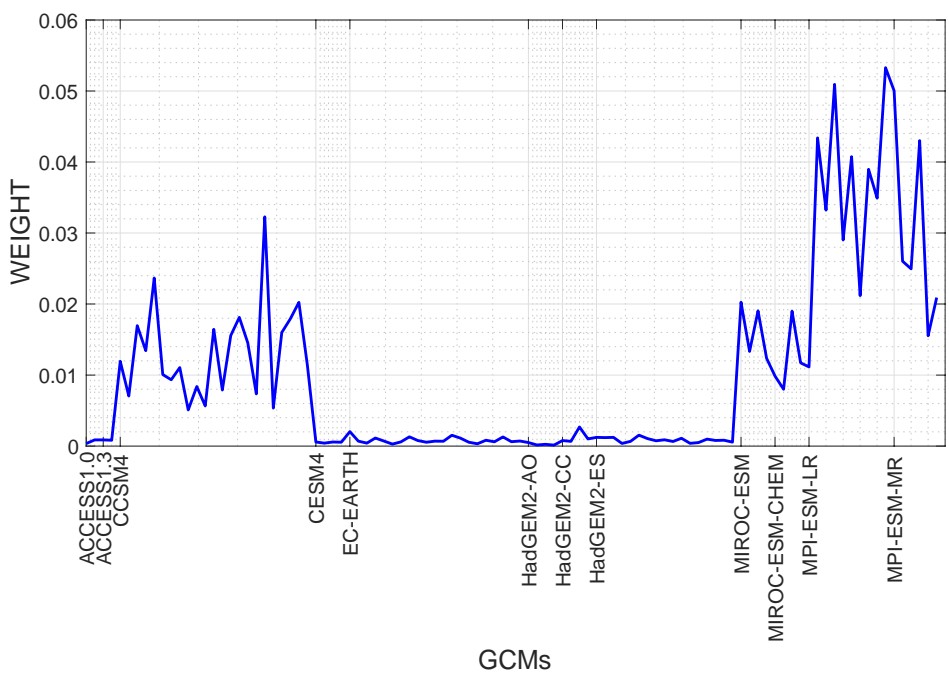

**Figure 6: Weight of each candidate model derived by AFTER-SSIM.**

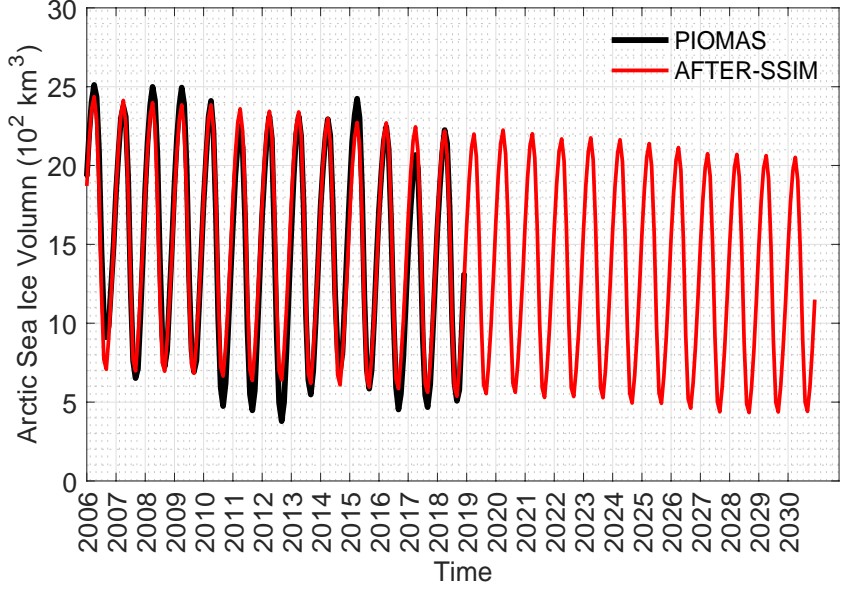




**Figure 7. Reconstruction and future projections for SIV based on AFTER-SSIM algorithm (The black line represents the PIOMAS data and the red line is the simulation of SIV based on the AFTER-SSIM, where the loss function has been optimised by a GA algorithm).**

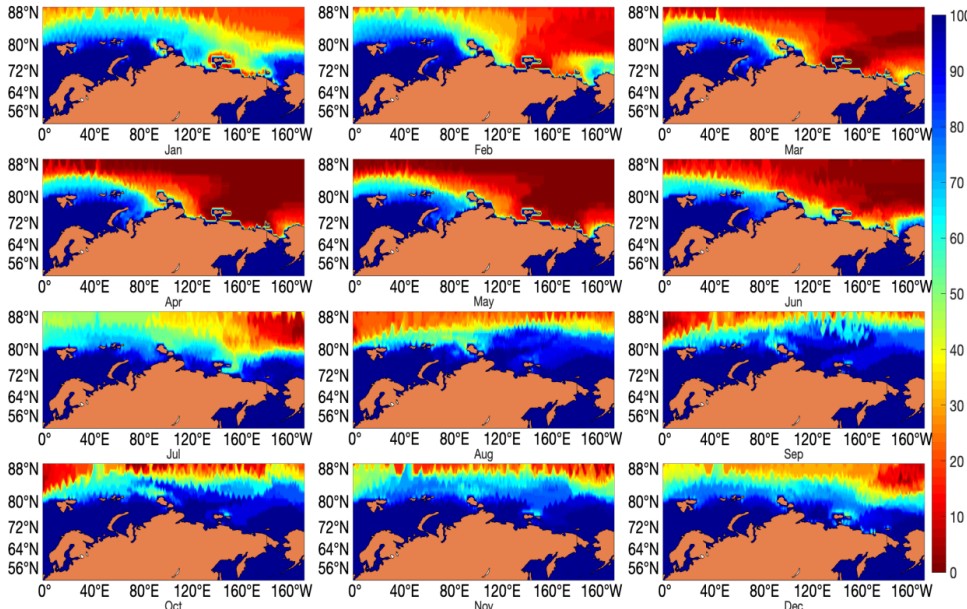

**Figure 8. Possibility of navigability for 1A super vessels on the Northern Sea Route for every month in the year of 2030 with 101 combined candidate realisations.**


**Code and Data availability**

**Code:**

Adaptive Forecasting Through Exponential Re-weighting algorithm:

Yang, Y., 2001: Combining forecasting procedures: Some theoretical results. Econom. Theory, 20, 176–222,

https://doi.org/10.1017/S0266466604201086.

Structural Similarity Index Measure algorithm:

Zhou, W., B. Alan Conrad, S. Hamid Rahim, and Simoncelli Eero P, 2004: Image quality assessment: from error

visibility to structural similarity. IEEE Trans Image Process, 13, 600–612.

**Data:**

ACCESS1.0, ACCESS1.3 datasets:

Bi, D., and Coauthors, 2013: The ACCESS coupled model: description, control climate and evaluation. Aust.

Meteorol. Oceanogr. J., https://doi.org/10.22499/2.6301.004.

CCSM4, CESM1 datasets:

Gent, P. R., and G. Danabasoglu, 2011: Response to Increasing Southern Hemisphere Winds in CCSM4. J. Clim.,

24, 4992–4998.

EC-EARTH datasets:

Fichefet, T., and M. A. M. Maqueda, 1999: Modelling the influence of snow accumulation and snow-ice formation

on the seasonal cycle of the Antarctic sea-ice cover. Clim. Dyn., 15, 251–268.

HadGEM2-ES, HadGEM2-CC, HadGEM2-AO datasets:

Mclaren, A. J., H. T. Banks, C. F. Durman, J. M. Gregory, and S. W. Laxon, 2006: Evaluation of the sea ice

simulation in a new coupled atmosphere-ocean climate model (HadGEM1). J. Geophys. Res., 111, C12014.

MIROC-ESM, MIROC-ESM-CHEM datasets:

Watanabe, M., M. Chikira, Y. Imada, and M. Kimoto, 2011: Convective control of ENSO simulated in MIROC.

J. Clim., 24, 543–562.

MPI-ESM-LR,MPI-ESM-MR datasets:

Notz, D., F. A. Haumann, H. Haak, J. H. Jungclaus, and J. Marotzke, 2013: Arctic sea-ice evolution as modeled

by Max Planck Institute for Meteorology's Earth system model. J. Adv. Model. Earth Syst., 5, 173–194.

PIOMAS datasets:





Zhang, J., and D. A. Rothrock, 2003: Modeling Global Sea Ice with a Thickness and Enthalpy Distribution Model

in     Generalized    Curvilinear    Coordinates.    Mon.    Weather    Rev.,    https://doi.org/10.1175/1520-

0493(2003)131<0845:mgsiwa>2.0.co;2.





**Author contribution**

| | | |
|---|---|---|
| | Wang Yangjun: | Methodology; Software; Writing- Original draft preparation |
| 485 | Liu Kefeng: | Writing - Reviewing and Editing |
| | Qian Longxia: | Data curation; Validation |
| | Zhang Ren: | Supervision; Conceptualization |



**Competing interests**

The authors declare that they have no known competing financial interests or personal relationships that could have appeared to influence the work reported in this paper.

The authors declare the following financial interests/personal relationships which may be considered as potential competing interests.