# Peer review of "Constraining Uncertainties in CMIP5 Projections of Arctic Sea Ice Volume with Observations"

_Ocean Science, 2020_

## Referee Comment (RC1) · Anonymous Referee #1 · 11 Aug 2020

This paper, os-2020-35 Constraining Uncertainties in CMIP5 Projections of Arctic Sea Ice Volume with Observations by Wang Yangjun, Liu Kefeng, Shan Yulong, and Zhang Ren, represents a valuable contribution to climate research and risk assessment communities. I anticipate that this method will be applied on CMIP6 models, and some different domains and variables (besides sea-ice cover in the high north). I would like to see this paper published in Ocean Science after minor revisions that must address the following key issues:

1) SIC from PIOMAS is not "reality", but an estimate, so please justify your choice of using PIMOAS SIC by showing how close is PIOMAS SIC to one or more well-

established remote sensing SIC products.

2) Calculate PIOMAS SIV on PIOMAS grid and then re-grid PIOMAS SIV to 1x1 deg.

3) Again, since PIOMAS is just a reanalysis (that actually uses NCEP Re1 – more than 20 years old atmospheric reanalysis system with pronounced surface biases in the Arctic), how would your conclusions change if you would use some other reanalysis product that provides SIT over the same period, e.g., ECMWF ORAS5. No need to do entire procedure again with ORAS5 or some other comparable product, but the authors should compare SIT and SIV fields (perhaps PIOMAS and ORAS5 are very close, if not that should be clearly indicated in the paper).

Below are some additional points that should be addressed:

Line 8: "... emission scenarios from 2006 to 2018, ..."

Line 10: "the genetic algorithm (GA)"? Could you briefly specify here (in few words or just a sentence) what is it.

Line 14: I would suggest: ".. possibility through the Arctic routes."

Line 20: The authors should also mention more updated, CMIP6 references for Arctic, e.g., SIMIP Community, 2020, Arctic sea ice in CMIP6, GRL, 46, e2019GL086749, https://doi.org/10.1029/2019GL086749.

Line 22: Perhaps you could cite here ("regional ecosystems (ref).") the following report: Chatman House, 2011, Arctic Opening: Opportunity and Risk in the High North (https://www.chathamhouse.org/publications/papers/view/182839).

Line 25: Now you can also mention CIMP6 results.

Line 26: I would suggest being more succinct here "available, showing a continued ..."

Line 34: What is "seasonal cycles of the September"? September state is a part of seasonal cycle. This needs to be clarified and/or rephrased.

Line 39: Typo: "this study tries to further assess the model performance of ..".

Line 43: It would be useful to put here some recent overview reference on multi-model benefits.

Line 47: I would suggest ". . . danger of overfitting and lead to . . .".

Line 71: Unit issue: Do you mean ".. less than a 0.1 m mean difference . . ."?

Line 75: Using PIMOS SIC actually takes you further away from reality. Hence, be specific how much does PIOMAS SIC deviates from remote sensing products like NSIDC or OSI SAF SIC?

Line 106: You mean ".. the SIC times SIT of each ...".

Related to equation (2), you should actually calculate SIV on the PIOMAS grid and then convert SIV to regular 1x1 deg grid. In general, it is best to perform as much as possible calculations on native grid before interpolation to some other grid.

Line 256: "well understood" is bit overconfident, perhaps I would suggest stating ".. have been adequately understood and represented . . ."

Line 258: You should add here: Notz, D., 2012, Challenges in simulating sea ice in Earth System Models. WIREs Clim Change, 3: 509-526. doi:10.1002/wcc.189

Acknowledgments: Are scripts used in this analysis publically available (and/or where they can be obtained)?

————————————————

---

## Referee Comment (RC2) · Anonymous Referee #2 · 21 Sep 2020

In this manuscript, the authors make use of an image-processing methodology for comparing Arctic sea ice properties (sea ice concentration, sea ice thickness, and sea ice volume) provided by several CMIP5 model outputs against the PIOMAS reanalyses. I am sorry for not being more positive at this stage, but in my opinion, the paper lacks scientific rigor. So, I do not see this manuscript ready for publication as a scientific paper. Please, see my arguments below:

1. First, I think the paper is out of the context of Ocean Science (OS). As shown in the journal's "Aims and scope" (https://www.ocean-science.net/about/aims_and_scope.html), OS covers the following fields:

[Figure]

- ocean physics (i.e. ocean structure, circulation, tides, and internal waves); - ocean chemistry; - biological oceanography; - air–sea interactions; - ocean models – physical, chemical, biological, and biochemical; - coastal and shelf edge processes; - paleoceanography.

It seems that the paper's subject (comparison of sea ice properties between model outputs and a reference reanalyses) matches better with another Copernicus journal, The Cryosphere (https://www.the-cryosphere.net/about/aims_and_scope.html) which cover the following aspects:

- ice sheets and glaciers; - planetary ice bodies; - permafrost and seasonally frozen ground; - seasonal snow cover; - sea ice; - river and lake ice; - remote sensing, numerical modeling, in situ and laboratory studies of the above and including studies of the interaction of the cryosphere with the rest of the climate system.

2, Besides what is claimed in the title (and throughout the text), the manuscript is not using "observations". Indeed PIOMAS has been largely used by the scientific community, but still, it is not an observational data set.

3, The text is sometimes confusing and it contains many vague statements. It needs to be substantially improved before publication. See just a few examples (non-exhaustive) below:

pg. 2, l33–35:Ăă"Wang & Overland, (2015) have reduced GCMs to 12 models by taking both the mean trends and seasonal cycles of the September sea ice extent (SSIE) projections from 1981 to 2005 into consideration."Ăă→ How and why?

pg. 2, l42–43:Ăă"Generally, there are two ways to improve the precision of projections; one is to find the model that best fits reality, while the other is to combine estimates with multi-models."Ăă→ Selecting the best performing model is not a way "to improve the precision of the projection" itself.

pg. 2, l51–53:Ăă"Additionally, the error method (e.g. relative error, root mean square

error), widely used in previous literature (Wang and Overland 2015a, 2009, 2012; Massonnet et al. 2011; Stroeve et al. 2012; Liu et al. 2013), cannot suitably fit the difference in spatial distribution (Zhou et al. 2004)"Âă→ this sentence is a bit confusing. Does a work from 2004 say that the methodology used by papers published in the 2010s is not adequate?

pg. 2, l51–53:Âă"sea ice thickness is also taken into account in the calculations." → Why "also"? What is the other variable that is taken into account? At this stage, the text referred only to sea ice thickness.

Sec. 2.1: The description of PIOMAS is very poor. What model is used? Is there data assimilation? How does it work? Which atmosphere forcing is used?

Etc.

I recommend that the authors make a careful review, sentence by sentence, of the manuscript.

4. The methodology is hard to follow. If I understood well, the methodology takes only into account the trivial trend and seasonal cycle(?) What about the interannual/decadal variability? Is it possible to predict "the possibility of navigability" without taking into account this important component of the sea ice variability?

5. Figures need to be substantially improved:

Fig. 1: It seems that the first panels are not needed at all. The authors didn't refer to them in the text. Also, the colormap doesn't make it easy to compare the plots.

Fig. 2: Very hard to distinguish the dates in the upper x-tick label; what does it mean "Year (o)"?; The y-label is wrong (10ˆ3 not 10ˆ2); What the authors call by "monthly change" was plotted only in the blue line but not in the red.

Fig. 3: y-ticklabel is missing; y-ticks are overlapping each other; the colormap and color bar range is not adequate.

These are comments only for the three first ones.

6, I am wondering why the authors didn't use the updated CMIP6 data since the files already available?